# Robustness and Complexity of Directed and Weighted Metabolic Hypergraphs

**DOI:** 10.3390/e25111537

**Published:** 2023-11-11

**Authors:** Pietro Traversa, Guilherme Ferraz de Arruda, Alexei Vazquez, Yamir Moreno

**Affiliations:** 1Institute for Biocomputation and Physics of Complex Systems (BIFI), University of Zaragoza, 50018 Zaragoza, Spain; 2Department of Theoretical Physics, University of Zaragoza, 50018 Zaragoza, Spain; 3CENTAI Institute, 10138 Turin, Italy; 4Nodes & Links Ltd., Salisbury House, Station Road, Cambridge CB1 2LA, UK

**Keywords:** hypergraphs, complexity, robustness, metabolism, communicability, search information

## Abstract

Metabolic networks are probably among the most challenging and important biological networks. Their study provides insight into how biological pathways work and how robust a specific organism is against an environment or therapy. Here, we propose a directed hypergraph with edge-dependent vertex weight as a novel framework to represent metabolic networks. This hypergraph-based representation captures higher-order interactions among metabolites and reactions, as well as the directionalities of reactions and stoichiometric weights, preserving all essential information. Within this framework, we propose the communicability and the search information as metrics to quantify the robustness and complexity of directed hypergraphs. We explore the implications of network directionality on these measures and illustrate a practical example by applying them to a small-scale *E. coli* core model. Additionally, we compare the robustness and the complexity of 30 different models of metabolism, connecting structural and biological properties. Our findings show that antibiotic resistance is associated with high structural robustness, while the complexity can distinguish between eukaryotic and prokaryotic organisms.

## 1. Introduction

A metabolic network [1,2,3,4,5] is a highly organized system of chemical reactions that occur in living organisms to sustain life and regulate cellular processes. Metabolic networks are incredibly complex because of the large number of reactions and the intricate web of interactions between molecules. Chemical reactions take some metabolites, usually called reactants or substrates, and turn them into products which can be used by other reactions. This complexity allows organisms to perform various functions and respond to various challenges, but it makes understanding them much more challenging. The key functions of metabolism are the production of energy, the conversion of food into building blocks of proteins, lipids, nucleic acids, and carbohydrates, and the elimination of metabolic wastes.

Given the network structure of metabolism, many researchers have attempted to characterize and understand it through network theory. It has been shown that graphs whose nodes are metabolites and are connected by chemical reactions have a scale-free distribution [3] and have been described as “among the most challenging biological networks and, arguably, the ones with most potential for immediate applicability” [6]. Other attempts have tried to give more concrete answers by focusing on graphs with reactions as nodes or bipartite graphs but missing a fundamental aspect of chemical reactions. To take place, they require a collective interaction of reactants to create multiple products. Hence, these are high-order interactions that graphs cannot fully capture. As network theory has advanced, new structures have been devised that can capture high-order interactions. These structures, called hypergraphs, have been very successful in fields such as social sciences [7,8,9,10,11,12,13,14,15,16,17], epidemiology [12,15,18,19,20,21], biology [22,23,24,25,26,27,28,29], etc. The potential of hypergraphs to describe cellular networks has been hypothesized in a perspective in 2009 [30]. Mapping into a hypergraph was also noted in [31,32,33,34], bringing attention to this new representation. Recently, Mulas et al. [35,36] applied hypergraphs to chemical networks, trying to capture the high-order nature of chemical reactions. In this paper, we take the concept of chemical hypergraphs and apply it to metabolic networks. In addition, we take it a step further by showing how including weights in the treatment allows no biological or structural information to be lost. Therefore, we argue that metabolic hypergraphs are the right framework to address and understand metabolism, allowing for a bridge between biology and network theory.

This article aims to lay the foundation for a theory of metabolic networks based on hypergraphs. We describe the method by which each metabolic network can be represented as a hypergraph and introduce two applicable measures, namely, communicability and search information.

The work is organized as follows. In Section 2, we give the mathematical definitions regarding metabolic hypergraphs. We also comment on previous studies in the field of metabolic networks and on how they can be viewed as a simplification of the metabolic hypergraph we propose here. In Section 3, we propose a generalization of communicability and search information for hypergraphs. We keep this section general enough so that these measures can be easily applied to any hypergraph, directed or undirected, weighted or not. We use metabolic hypergraphs as an example, and we report the results in Section 4. We conclude by commenting on the possibility that this framework offers of motivating further research in this area.

## 2. Metabolic Networks as Hypergraphs

In this section, we give a formal definition of metabolic hypergraphs and introduce the notation that is used to characterize them.

### 2.1. Hypergraphs Definition

A hypergraph $H=\left\{V,E\right\}$ is a set of vertices or nodes v∈V and hyperedges e∈E. Each hyperedge is a subset of *V* such that different nodes interact with each other if and only if they belong to the same hyperedge. Thus, unlike traditional graphs, where edges connect pairs of nodes, hyperedges represent interactions involving multiple nodes. If the dimension |e| of the hyperedges is 2, then the hypergraph is equivalent to a conventional graph. The total number of vertices is denoted as N=|V| and the number of hyperedges as M=|E|.

To interpret metabolic networks as hypergraphs, we first need to define a special type of hypergraph introduced by Chitra et al. [37]. A hypergraph with edge-dependent vertex weights (EDVW) $H=\left\{V,E,W,\Gamma \right\}$ is a set of vertices or nodes v∈V, hyperedges e∈E, edge weights w(e), and edge-dependent vertex weight $\gamma_{e}\left(v\right)$. If $\gamma_{e}\left(v\right)=\gamma \left(v\right)\;\forall \;e\in E$, then the hypergraph is said to have edge-independent vertex weight. All the weights are assumed to be positive. These types of weights are a unique property of some higher-order systems and are crucial for encoding in the hypergraph all the information contained in metabolic networks.

In this paper, we deal with directed hypergraphs, which are an extension of directed graphs. In a directed hypergraph, each hyperedge is associated with a direction similar to the direction of an arrow connecting two vertices in a directed graph. In this context, a hyperedge $e_{j}$ is divided into a head set $H(e_{j})$ and a tail set $T(e_{j})$. Similarly to the arrow, the direction goes from the tail to the head set, with the difference that the directed hyperedge is connecting multiple vertices. A vertex can belong solely to either the head or the tail of a hyperedge, but not both. Unless explicitly stated otherwise, any hypergraph in this paper is considered to be directed.

Additionally, we define $k_{v}^{out}$, the out-degree of a vertex v∈V, as the number of hyperedge-tails that include *v*. Similarly, $k_{v}^{in}$ denotes the in-degree of a vertex v∈V, the number of hyperedge-heads in which *v* is contained. We also use |H(e)| and |T(e)| to represent the number of vertices belonging to H(e) and T(e), respectively.

Given a directed hypergraph H = $\left\{V,E\right\}$ of *N* vertices and *M* hyperedges, the incidence matrix is the matrix $I\in R^{N\times M}$ such that:(1)$$I_{ij}=\left\{\begin{array}{cc} 1 & \;\mathrm{if}\;v_{i}\in H\left(e_{j}\right)\\ -1 & \;\mathrm{if}\;v_{i}\in T\left(e_{j}\right)\\ 0 & \;\mathrm{if}\;v_{i}\notin e_{j} \end{array}\right.,$$
where $H(e_{j})$ and $T(e_{j})$ are, respectively, the heads and the tails of the hyperedges $e_{j}$. We can rewrite the incidence matrix as
(2)$$I=I_{H}-I_{T},$$
where we separate the contributions coming from the head ($I_{H}$) and the tail ($I_{T}$) of the hyperedges in order to work with positive signed matrices. It is useful to mathematically define sinks and sources. A source is a node (or hyperedge) that has zero in-degree (or empty tail) and non-zero out-degree (or empty head). A sink is a node (or hyperedge) that has non-zero in-degree (or empty tail) and zero out-degree (or empty head).

### 2.2. Metabolic Hypergraphs

In this article, we focus on metabolic networks. A metabolic network [2] is a set of biological processes that determines the properties of a cell. Several reactions are involved in metabolism, grouped into various metabolic pathways. A metabolic pathway is an ordered chain of reactions in which metabolites are converted into other metabolites or energy. For example, the glycolysis pathway is the set of reactions involved in the transformation of one molecule of glucose into two molecules of pyruvate, producing energy. Metabolic networks are among the most challenging and highest-potential biological networks [3,6]. The way to represent a metabolic network on a graph is not unique, and several approaches have been tried. One possible way is to consider metabolites (or reactions) as nodes and connect them if and only if they share a reaction (or metabolite). The resulting graph is undirected, and this may change the structural properties of the network in an undesirable way. In [38], the authors analyze the same dataset that we analyze for *E. coli* and propose a directed graph with reactions as nodes that take into account the directionality of the reactions, highlighting the difference with the undirected counterparts.

However, reactions are intrinsically higher-order interactions since they can occur only when all reactants are present. In Figure 1, we illustrate the way to map a chemical reaction network into a hypergraph. The resulting hypergraph is a directed hypergraph with edge-dependent vertex weight, which we will refer to as a metabolic hypergraph for brevity. More formally, we define a metabolic hypergraph as a 3-tuple $H=\left\{V,E,S\right\}$, where $V=\{v_{1},v_{2},\cdots v_{N}\}$ is a set of N metabolites (vertices) and E is a set of oriented reactions (hyperedges). Each e∈E is a pair (T(e),H(e)), the tail and the head of the hyperedge which correspond, respectively, to the inputs and outputs of the reaction. Note that T(e) or H(e) can also be empty sets. This is the case for external reactions that introduce inside the cell the ingested metabolites (the tail is an empty set) and external reactions that secrete metabolites (the head is an empty set). We also call the former source reactions and the latter sink reactions, and their effect on the measurements is discussed in more detail in Section 3. S is the stoichiometry matrix associated with the chemical network, and it represents the EDVW of the hypergraph. Indeed, one can notice that *S* can be rewritten using the EDVW matrix Γ as S=Γ∘I, where I is the directed incidence matrix and “∘” is the element-wise matrix product.

### 2.3. Literature Background

There are different techniques for studying metabolic networks. Popular methods employ kinetic metabolic models [39,40] and stochastic chemical kinetics [41] to study the dynamics of metabolites concentrations in metabolism. While these models are crucial to understanding the complex dynamics of metabolic networks, they require the notion of the kinetic rates constant, the rates at which metabolites are consumed per reaction, which are usually not available [42]. What instead is generally known are the reactions, the stoichiometry coefficient, and the structure of the metabolic network. Thus, several graph representations of metabolic networks have been tried. The most common one is the reaction adjacency matrix (RAG) defined as $A^{\mathrm{RAG}}={\hat{S}}^{T}\hat{S}$ [32,38], where $\hat{S}$ is the boolean version of the stoichiometry matrix. The biggest limitation of this model is that is undirected, while we know that the direction of reactions is chemically very important. A big improvement was proposed in [38], where the authors proposed a flux-dependent graph model that accounts suitably for the directness of the reactions. However, graph representations of these systems are still missing a crucial point, which is the fact that reactions are higher-order objects which involve the interactions of all input metabolites to produce output metabolites. Therefore, hyperedges are the natural mathematical object for encoding reactions. Mulas et al. [36] already took a step in this direction by defining a Laplace operator for chemical hypergraphs. The last step we make is to incorporate into the hypergraph model the weights associated with metabolites and reactions, using a similar framework to the EDVW defined in [37]. In this last modification to the model, it is crucial to include biological and chemical constraints in the model.

The main advantage of the metabolic hypergraph framework we propose is that it captures all the physical properties that a metabolic network displays: the directness of reactions, the higher-order interactions, and the chemical properties like mass conservation, due to the inclusion of weights. This framework represents a link between network theory and biology. Another commonly employed method for analyzing large-scale metabolic network models is constraint-based metabolic modeling, such as flux balance analysis. Flux balance analysis (FBA) is used to obtain steady-state reaction rates that are consistent with a metabolic network and linear constraints on the reaction rates, without necessitating any knowledge about the kinetic parameters [43]. FBA is a method of finding steady-state solutions [44,45], yet, one needs to perform additional analyses to determine the relevance of each reaction or metabolite to the solutions obtained. For this scope, hypergraph theory provides a lot of tools that could be used alongside FBA.

We remark that the previous graph representation of metabolic networks can be seen as a pairwise projection of a metabolic hypergraph. For example, the RAG is an undirected projection of the hypergraph, as in [46], and the flux-dependent graph [38] is similar to the normalized adjacency matrix defined in [47] but extends to directed and weighted hypergraphs. Projections are a pairwise simplification and can perform well depending on the task, but they do not contain all the information. For example, in [32], the authors start with a hypergraph formalism and project it to a reaction adjacency to evaluate the number of extreme pathways in metabolic networks.

### 2.4. Dataset

In our experiments, metabolic hypergraphs are generated from the stoichiometry matrix of the models stored in the BiGG database [48]. We analyze 30 different models, with an increasing number of nodes describing different organisms (see Table A1 in Appendix A for the exact number of nodes and reactions of each BiGG model). We chose the metabolic networks in order to have a reasonable variety of organisms, and we avoided very large networks because of the computational costs. The majority of the data are composed of bacteria that can be divided into classes like antibiotic-resistant, aerobic or anaerobic, Gram-positive or Gram-negative. The other organisms are eukaryotes, and one is in the Archaea domain. All data are publicly available on the BiGG models web page [49] in different formats. In this analysis, the *.json* format is used. The data contain information on metabolites, reactions, and genes. Metabolites are identified by a Bigg ID, consisting of an abbreviation defining their type, for example, “h” for hydrogen and “ATP” for adenosine triphosphate, and a subscript indicating the compartment to which they belong. Regarding the reactions, in addition to their IDs, the metabolites belonging to them are given, with their respective stoichiometric coefficients. We work in the convention in which a metabolite with a positive stoichiometric coefficient is a product; otherwise, it is a reactant. In the BiGG dataset, the direction of the reactions is also determined using the parameters “lower_bound” and “upper_bound”. These parameters are associated with each reaction and correspond to the maximal flux of metabolites that can flow through. Values of lower_bound =0 and upper_bound >0 mean that the reaction is annotated correctly, following the convention. On the contrary, if lower_bound <0 and upper_bound =0, the reactions are written with inverted orientations. These two parameters combined also determine if a reaction is reversible or not. If a reaction is reversible, both the direct and inverse reactions are present and will be characterized by a lower_bound <0 and upper_bound >0. We recall that we treat reversible reactions as two distinct hyperedges, see Figure 1 for a visual example. It is important to notice that few reactions have lower_bound =0 and upper_bound =0. In practice, this implies that no flux of metabolites can flow through, so those reactions are discarded. The origin of the reaction bounds depends on the BiGG models considered. For example, both models for *Mycobacterium tuberculosis H37Rv* have some reactions with lower_bound =0 and upper_bound =0 identified via flux variability analysis (FVA). In the case of the *Synechococcus elongatus PCC 7942*, the bounds are obtained experimentally. We decide to proceed with this convention, but using relaxed reaction boundaries to include these reactions is also a valid option.

All the metabolites present in the BiGG models were kept; we did not discard dead-end metabolites.

Lastly, we highlight that some hyperedges may have an empty tail or head. These hyperedges correspond to reactions involved in the transportation of metabolites from the outside of the cell to the inside or vice versa. For example, EX_h2o_e (H${}_{2}$O exchange) is the reaction that takes the water from the environment and brings it into the metabolism. The metabolites outside the metabolism are not present in the BiGG models, and for this reason, the reaction appears as “ → h2o_e”, with an empty tail. Therefore, sometimes they may represent sinks and sources in the hypergraph. By source, we mean a node or hyperedge from which you can start and leave but never go back, while a sink is a trapping node or hyperedge that, if it is reached, is impossible to leave.

## 3. Measurements

In this section, we define two measures of the chemical hypergraph based on the notion of paths or walks on hypergraphs. A *walk* of length *l* from node $v_{0}$ to node $v_{l}$ is defined as a sequence of alternating nodes and hyperedges $\left(v_{0},e_{1},v_{1},e_{2},v_{2},...e_{l},v_{l}\right)$. We also define the *dual walk* from hyperedge $e_{0}$ to hyperedge $e_{l}$ of length *l* as the alternating sequence of alternating nodes and hyperedges $\left(e_{0},v_{1},e_{1},v_{2},e_{2},...v_{l},e_{l}\right)$. We are interested in both metabolites and reactions, which is why it is useful also to consider the dual walk.

### 3.1. Hypergraph Communicability

We are usually interested in understanding how paths are distributed because that is how information and interactions spread. In social systems, for example, the more paths connecting two nodes, the easier is for information to spread from one to another. Also, if one path of connection fails, the information can still be spread through other paths, even if they are longer than the path that failed. For this reason, the notion of paths and communication between nodes can also be related to the robustness of the network. However, having a robust network is not always positive. The same reasoning about the spreading of information applies to the spreading of viruses. If a network is robust, it is way more difficult to design containment strategies for the virus, since shutting down a connection might not be enough because of the presence of alternative paths. A way to measure how nodes communicate within a network is called communicability, and we extend this definition to hypergraphs.

The communicability [50,51] between two pairs of nodes *p* and *q* is defined as the weighted sum of all walks starting from node *p* and ending at node *q*, as in
(3)$$G_{pq}=\sum_{k=0}^{\infty }c_{k}n_{pq}^{k},$$
where $n_{pq}^{k}$ is the number of walks from *p* to *q* and $c_{k}$ is the penalization for long paths. The most common choice is $c_{k}=\frac{1}{k!}$ so that you recover an exponential expansion. For a graph, $n_{pq}^{k}$ can be easily found by taking the k-power of the adjacency matrix, ${\left(A^{k}\right)}_{pq}$. Hypergraphs do not have a unique definition of adjacency matrix; we thus have to use the definition of walk given above. The vertex-to-vertex communicability for a hypergraph with incidence matrix I is defined as
(4)$$G_{pq}^{V}=\sum_{k=0}^{\infty }\frac{{\left({\left(I_{T}I_{H}^{t}\right)}^{k}\right)}_{pq}}{k!},$$
or, in matrix form,
(5)$$G^{V}=e^{I_{T}I_{H}^{t}},$$
where ${\left(\cdot \right)}^{t}$ indicates the transpose of the matrix. In metabolic hypergraphs, we are also interested in how reactions communicate with each other. For this reason, we define the hyperedge-to-hyperedge communicability based on the notion of dual path
(6)$$G_{pq}^{E}=\sum_{k=0}^{\infty }\frac{{\left({\left(I_{H}^{t}I_{T}\right)}^{k}\right)}_{pq}}{k!},$$
or, in matrix form,
(7)$$G^{E}=e^{I_{H}^{t}I_{T}}.$$

The Estrada index [50,52] of a hypergraph *H* is generalized as
(8)$$\begin{array}{c} EE^{V}\left(H\right)=\mathrm{Trace}\left(G^{V}\right),\\ EE^{E}\left(H\right)=\mathrm{Trace}\left(G^{E}\right). \end{array}$$

One can notice that the matrices $I_{T}I_{H}^{t}$ and $I_{H}^{t}I_{T}$ have the same spectrum except for the number of zero eigenvalues because of the difference in size. This means that for M>N, for example (which is usually the case in metabolic hypergraphs), the Estrada index defined on nodes and the one defined on the hyperedges are related by $EE^{E}\left(H\right)=EE^{V}\left(H\right)+\left(M-N\right)$. We use the Estrada index defined on the nodes to measure the hypergraph robustness, also known as natural connectivity, as
(9)$${\bar{\lambda }}^{V}=\log\left(\frac{EE{\left(H\right)}^{V}}{N}\right).$$

The same definition holds for ${\bar{\lambda }}^{E}$ with the proper normalization.

Since computing the exponential of very large matrices might be a difficult numerical task, we use an approximation for the calculation of the robustness based on eigenvalue decomposition. For simplicity, let us call $A^{V}=I_{T}I_{H}^{t}$ (the same reasoning holds for $A^{E}=I_{H}^{t}I_{T}^{t}$) and order the spectrum of $A^{V}$ in such a way that $\lambda_{1}>\lambda_{2}>\lambda_{3}>...\lambda_{N}$. Then, the natural connectivity or robustness of the hypergraph becomes
$$\begin{array}{cc} {\bar{\lambda }}^{V} & =\log\left(\sum_{i=1}^{N}e^{\lambda_{i}}\right)-\log\left(N\right)\\ \; & =\log\left[e^{\lambda_{1}}\left(1+\sum_{i=2}^{N}e^{\lambda_{i}-\lambda_{1}}\right)\right]-\log\left(N\right)\\ \; & =\lambda_{1}+\log\left(1+\sum_{i=2}^{N}e^{\lambda_{i}-\lambda_{1}}\right)-\log\left(N\right)\\ \; & =\lambda_{1}-\log\left(N\right)+O\left(e^{-(\lambda_{1}-\lambda_{2})}\right). \end{array}$$

Thus, if the spectral gap is large enough, the natural connectivity is dominated by the largest eigenvalue. Since the correction is exponential, this approximation is usually quite good. As a consequence of the common spectrum of $I_{H}^{t}I_{T}$ and $I_{T}I_{H}^{t}$, the difference in robustness is approximately ${\bar{\lambda }}^{V}-{\bar{\lambda }}^{E}\approx \log\left(\frac{M}{N}\right)$, which is usually quite small. It is worth noting that the largest eigenvalue scales with the system size, i.e., with the number of nodes and hyperedges. The normalization factor −log(N) mitigates this scaling effect, but a partial correlation is still expected. This correlation was present in the original graph definition of natural connectivity [53]. This could be a problem when comparing systems with very different scales. For this reason, in this paper, we are comparing hypergraphs with a similar system size.

This generalization of communicability applies also to undirected hypergraphs by substituting $I_{H}$ and $I_{T}$ with the undirected incidence matrix *I*.

### 3.2. Hypergraph Search Information

Rosvall et al. [54,55] introduced the concept of search information, as a measure of complexity in urban graphs. The idea is to measure the number of binary questions one has to make in order to locate the shortest path connecting a node *s* to a node *t*. As a consequence, this measure is based on walks like the communicability, but with the crucial difference that it considers only the shortest paths. This allows us to link the search information with the notion of complexity. While alternative pathways tend to make the network more robust, they also make the probability of finding the shortest path decrease and the complexity increase. This trade-off is the reason that motivated us to consider communicability and search information together.

In [54], the search information is defined as a matrix *S* with entries
(10)$$S{\left(i,j\right)}^{V}=-\log_{2}\left(\sum_{\left\{p(i,j)\right\}}P\left(p(i,j)\right)\right),$$
where $\left\{p(v_{i},v_{j})\right\}$ is the set of all shortest paths from node $v_{i}$ to node $v_{j}$.

The original definition was made for undirected and unweighted ordinary graphs, so a very different structure from directed hypergraphs with edge-dependent vertex weight, but the meaning remains the same. What changes is the probability of following the shortest path. The probability of making a step is proportional to the stoichiometric coefficients of the starting and arriving nodes, similar to what has been performed in the normalized flow graph in [38]. The probability of taking a step in a directed hypergraph with EDVW is
(11)$$\begin{array}{c} P\left(v\overset{}{\to }e\right)=\frac{\gamma_{e}\left(v\right)}{\sum_{h}\gamma_{h}\left(v\right)},\\ P\left(e\overset{}{\to }v\right)=\frac{\gamma_{e}\left(v\right)}{\sum_{n}\gamma_{e}\left(n\right)}. \end{array}$$

The probability of following a path is derived via multiplication of the single-step probability,
(12)$$P\left(v_{0},v_{l}\right)=P\left(v_{0}\overset{}{\to }e_{1}\right)P\left(e_{1}\overset{}{\to }v_{1}\right)\cdots P\left(e_{l}\overset{}{\to }v_{l}\right).$$

It is important to note that the search information might be ill defined if the hypergraph has sources or sinks. For example, by definition, there are no paths from a sink node $v_{\sin k}$ to any other nodes *v*, making the definition of $S(v_{\sin k},v)$ unclear in this case. What we do to solve the problem is to set $S(v_{\sin k},v)=0$ and then not count sink and source nodes when computing the average. With this convention, the access, hide, and average search information are defined as
(13)$$\begin{array}{cc} \; & A^{V}\left(s\right)=\frac{1}{N-N_{\mathrm{sources}}}\sum_{t}S^{V}\left(s,t\right)\\ \; & H^{V}\left(t\right)=\frac{1}{N-N_{\sin\mathrm{ks}}}\sum_{s}S^{V}\left(s,t\right)\\ \; & {\bar{S}}^{V}=\frac{1}{(N-N_{\sin\mathrm{ks})}\left(N-N_{\mathrm{sources}}\right)}\sum_{s,t}S^{V}\left(s,t\right). \end{array}$$

As a consequence, the access information of a sink and the hide information of a source will be set to zero. Following [54], we introduce an additional normalization factor $\log_{2}N$ to take into account size effects. With this additional term, we did not observe any correlation between the average search information and the number of metabolites or reactions. We denote the normalized average search information as $\sigma^{V}=\frac{{\bar{S}}^{V}}{\log_{2}N}$. The interpretation of these measures is very intuitive. The access information measures how easy it is to reach the other nodes in the network, while the hide information estimates how hidden a node is. Consequently, very central and connected nodes in the hypergraph have low hide information because there are a lot of paths leading to them, but they have relatively high access information because there are also many paths departing from such nodes.

## 4. Results and Discussion

In this section, we apply the previously defined metrics to a range of metabolic hypergraphs. As illustrated in Figure 1, these hypergraphs were constructed by starting with metabolic networks obtained from the BiGG dataset [48]. The metabolic networks were selected to have a reasonable variety of organisms. The primary goal of this section is to demonstrate the practical application of our framework and the defined measurements.

### 4.1. Exploring the E. coli Core Model: A Practical Example

To provide a tangible illustration of our methodology, we focus on the BiGG model known as e_coli_core [56]. This model represents a small-scale version of Escherichia coli str. K-12 substr. MG1655, making it an ideal candidate for demonstrating the performance of our metrics and understanding their limitations. Additionally, an Escher map for this model is available online [57].

In Figure 2, we show the access vs. hide information for reactions and metabolites. Regarding the reactions (Figure 2a), the measure correctly identifies the Biomass reaction as a central hub. Reactions are plotted with different colors based on the biological pathway they belong to. We can clearly see the behavior of sinks and sources in the reactions belonging to the extracellular exchange pathway. The pathways do not tend to separate into clusters, indicating that they all have a similar complexity. This could be an effect of the simplicity of this model or could be a property shared by all organisms. We did not investigate further since the scope of this section was just to provide a practical example, but it could be worth it to explore it in future work.

We also comment on the reactions that are ranked the highest according to average communicability. The average communicability is defined as ${\bar{G}}_{e}=\frac{1}{M}\sum_{h\in E}G_{he}^{E}$ and is shown in Figure 3. Notably, the Biomass reaction (first highest average communicability) and ATP synthase (second highest average communicability) are correctly identified as central reactions within the metabolism. The Biomass reaction is responsible for cell growth, while ATP synthase plays a crucial role in ATP synthesis, the primary energy source for the organism. The production of ATP is mainly due to the consumption of oxygen that occurs through the reaction CYTBD (cytochrome oxidase bd—sixth highest average communicability). When oxygen is unavailable, Escherichia coli can still survive due to the activation of the anaerobic pathway, which derives energy from the reaction THD2 (NAD(P) transhydrogenase—third highest average communicability).

Regarding the metabolites (Figure 2b), we observe a clear distinction between those belonging to the cytosol compartment and those located in the extracellular compartment. As expected, extracellular metabolites tend to have, on average, higher hide information. It is important to clarify that metabolites with zero hide information are those that remain initialized to zero because they are unreachable. However, an instructive observation could be made on o2_c. As commented in Section 3, a node with low but non-zero hide information is expected to be a central hub, but in reality, it has a very low degree. The explanation for this helps us to understand the implications of network directionality. The node o2_c is only connected to the core metabolism via the irreversible CYTBD reaction as a substrate. Consequently, there cannot be any directed path from the core metabolism to o2_c, only the opposite. We conclude that the node o2_c does not belong to the largest strongly connected component. In practice, it behaves very similarly to a source node. Nonetheless, the hide information is not zero because a pathway originates from the transport of external oxygen to the cytosol. In contrast, in cyanobacteria, algae, and plants (not investigated here), O_2_ is produced via oxygenic photosynthesis. In those organisms, O_2_ should be part of the strongly connected component.

### 4.2. Robustness and Complexity across Organisms

Our study assesses the robustness and complexity of 30 distinct metabolic hypergraphs derived from various eukaryotic and prokaryotic organisms. We selected the models to ensure a good range of diversity while avoiding having too many models for a single organism. For example, *Escherichia coli* has over 50 BiGG models. Analyzing all of them could be intriguing as well, but our primary interest lies in comparing organisms rather than models. Additionally, it is worth noting that certain BiGG models exhibit very large metabolisms, featuring thousands of metabolites and reactions. While these large models do hold potential relevance within the scope of our paper, the significant size of the corresponding metabolic networks renders the computational cost of the search information high. It is possible to study a large metabolic network individually, but the cost of comparing many together is prohibitive. To maintain computational tractability, we restrict our analysis to metabolic networks with no more than 2000 nodes or reactions.

Assessing the robustness of metabolic networks is an important task, and many definitions exist [31,34]. Here, we use natural connectivity to evaluate the network structure’s robustness. In Figure 4, we present the computed robustness values for several organisms arranged in ascending order. The BiGG models associated with the organisms *Staphylococcus aureus subsp aureus* [58,59], *Mycobacterium tuberculosis* [60,61], *Acinetobacter baumannii AYE* [62], and *Salmonella enterica* [63] are represented in different colors because they are bacteria that have evolved resistance to antibiotics. Except for the first *Staphylococcus aureus subsp aureus* model, antibiotic-resistant bacteria tend to exhibit relatively high robustness compared to other organisms. We measured the Spearman’s rank correlation between robustness and antibiotic resistance, obtaining a value of 0.424, revealing a moderate correlation. Here, the definition of robustness is based on the network’s resilience to random or targeted node removal. The concept of natural connectivity quantifies this resilience by counting the number of closed loops in the network. If there are many alternative paths, it is less probable that node removal will disconnect the network. In the context of biology, antibiotics operate by targeting and inhibiting some specific reactions, without which the cell dies [1]. Therefore, having a structurally robust metabolism is advantageous as it allows the organism to circumvent antibiotic inhibition by utilizing alternative reactions or pathways. However, this is not the whole picture since many other factors play a role. For example, bacteria are naturally subjected to random mutations that may strengthen their response to antibiotics, and this may not necessarily be reflected in a high structural hypergraph robustness. Conversely, a very robust metabolic hypergraph, with many alternative paths, may have a few but very important reactions that are easy to target with antibiotics. Hence, high structural hypergraph robustness does not guarantee antibiotic resistance.

The complexity of metabolic networks is anticipated to be quite similar across organisms since they share many common reactions and metabolic pathways. Nevertheless, some differences are expected in the metabolism of aerobic and anaerobic organisms, as well as between eukaryotes and prokaryotes. Aerobic and anaerobic organisms should have a different metabolism because of the different ways they produce energy, while eukaryotes and prokaryotes have significantly different cell structures. With this in mind, we measure the average search information of the 30 different metabolic hypergraphs and report the results in Figure 5. We notice a clear separation between eukaryotes and some aerobic organisms, showing a high complexity, and prokaryotes, which have a lower complexity. A few outliers exist, including *Staphylococcus aureus subsp aureus N315*, which exhibits high complexity, potentially due to unusually large weights associated with certain reactions compared to other organisms. Setting all the weights to 1 would indeed lead to a much lower complexity, ranked slightly below the average, indicating a possible bias. In addition, one can also notice that the other model for *Staphylococcus* has a low complexity. Another outlier is the first model we analyzed for *Homo sapiens—erythrocytes* [64] that may be expected to be complex. However, it is important to note that this model refers just to the erythrocyte metabolism (blood cells) rather than the entire human metabolism. Erythrocytes lack mitochondria and produce ATP through anaerobic glycolysis, so their metabolism could be closer to that of anaerobic organisms. Conversely, the low complexity of the aerobic organisms *Acinetobacter baumannii AYE*, *Pseudomonas putida*, and *Helicobacter pylori* is curious, and we do not have a clear motivation. Note that a generic human (*Homo sapiens*) cell has a similar complexity to a yeast cell (*Saccharomyces cerevisiae*). That is expected. Eukaryote cells have similar metabolic pathways. The additional complexity in human metabolism is due to multi-cellularity, which is not accounted for in this study.

## 5. Conclusions

Metabolic networks are very large and complex systems. For this reason, it is important to build a framework able to unite biology and network theory. Many successful studies have represented metabolic networks as graphs with metabolites as nodes, reactions as nodes, or both. Taking a step further, with the employment of hypergraphs, we are able to capture what all of these previous graph representations were missing, the higher-order interactions of reactions. In this paper, we show how metabolic networks are naturally mapped into hypergraphs. In particular, the stoichiometry matrix can be viewed as a weighted incidence matrix of a directed hypergraph with edge-dependent vertex weight. No information is lost representing metabolic networks as hypergraphs: the higher-order interactions between metabolites, the directionalities of reactions, and the stoichiometric weights are all included.

Within this novel framework, we propose two measurements to characterize a hypergraph’s robustness and complexity. We apply them to directed hypergraphs with EDVW, but the generalization to undirected and unweighted hypergraphs is straightforward. This approach allows for analysis at the local scale, with communicability and access and hide information, and at the global scale, with natural connectivity as a measure of robustness and average search information as a measure of complexity. We comment on the complications introduced by directionality and how they can be reflected in the measures. To illustrate the practical application of our framework and metrics, we present an example using the e_coli_core model. This small-scale metabolism demonstrates how our metrics operate locally, and it offers valuable insights into the behavior of metabolic hypergraphs. At the global scale, we compare 30 different BiGG models in robustness and complexity, leading to some interesting results. We show that the metabolisms of organisms that have evolved resistance to antibiotics are associated with hypergraphs that display high robustness. Furthermore, we observe that eukaryotic and prokaryotic organisms have different complexity values.

In our analysis of complexity, we excluded the source and sink reactions because they create problems when computing the search information (they are unreachable hyperedges). Another possibility could be to add a boundary node, representing the environment around the cell, that links the sinks with the sources. In this way, the search information is no longer ill defined and the external reactions could be included in the analysis. However, the introduction of such externally may have undesired effects on the measures, like introducing new and biologically unmotivated shortest paths. It is worth mentioning that an additional boundary node could be crucial when incorporating hypergraph dynamics in the model. A possibility for future works could be modifying the definition of the average search information and the probability of taking a step in the hypergraph. Here, we consider a walk biased by the stoichiometric weights, but more options could be explored. One possibility is to define the probabilities based on the communicability measure or on the rates computed via flux balance analysis [38,43]. Indeed, from flux balance analysis, we obtain rates that could be interpreted as edge-dependent vertex weights, substituting the stoichiometric coefficients. The union of FBA with hypergraph theory, to the best of our knowledge, has not been studied yet and could be an original contribution to the field. Also, we did not consider the information regarding genes that are contained in the BiGG models. Genomics plays a crucial role, especially in resistance to antibiotics, and for this reason, it could be interesting to integrate it into this framework. Another possibility is to apply our measures to other contexts, like social or technological hypergraphs.

We believe that this framework represents a promising approach to bridging network theory and biology. We hope that it may serve as a starting point, potentially reaching experts in the field who could further refine and utilize these findings to obtain more biological insights.

## Figures and Tables

**Figure 1 entropy-25-01537-f001:**
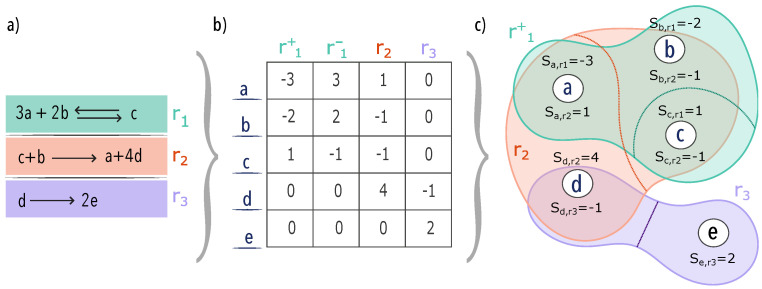
An example of a metabolic network mapped into a hypergraph with edge-dependent vertex weight. In (**a**), we present a small network composed of three reactions and five metabolites. The first reaction $r_{1}$ is reversible and is represented with the double arrow. In (**b**), we show the corresponding stoichiometry matrix. Reactants are negative and products are positive. Note that we need to split the reversible reaction into two irreversible reactions $r_{1}^{+}$ and $r_{1}^{-}$ to write it in matrix form. This stoichiometry matrix is the weighted incidence matrix of the hypergraph with edge-dependent vertex weights shown in (**c**). For the sake of visualization, only the hyperedge $r_{1}^{+}$ is shown. The hyperedge $r_{1}^{-}$ is just the same but with the opposite sign. Note that weights are both positive and negative, meaning that the hypergraph is directed. Indeed, we separate the head and tail of each hyperedge with a dashed line.

**Figure 2 entropy-25-01537-f002:**
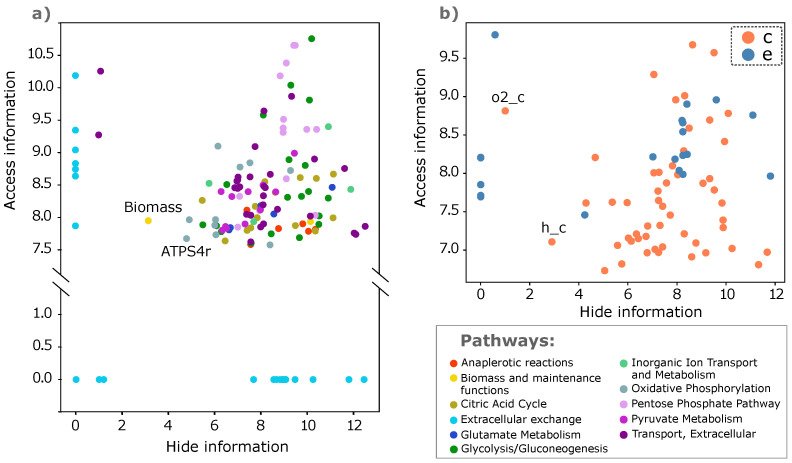
Access vs. hide information for reactions (**a**) and metabolites (**b**). Reactions are colored differently according to the pathway they belong to. Note that the *y* axis is cut for visualization purposes. Metabolites are divided into compartments; *c* stands for cytosol compartment and *e* for extracellular space.

**Figure 3 entropy-25-01537-f003:**
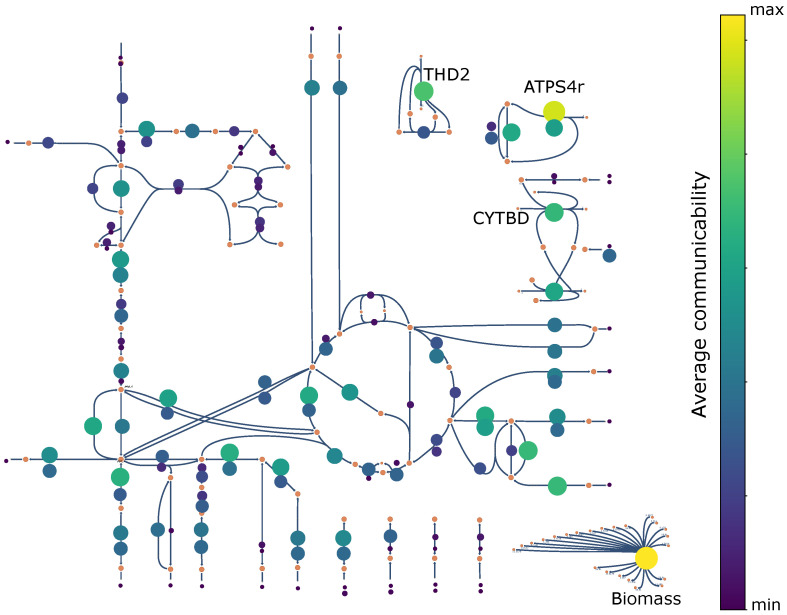
Reactions’ average communicability for the e_coli_core model. A simplified Escher map is used as a background to help with the visualization. For a more accurate version of the map, visit [57].

**Figure 4 entropy-25-01537-f004:**
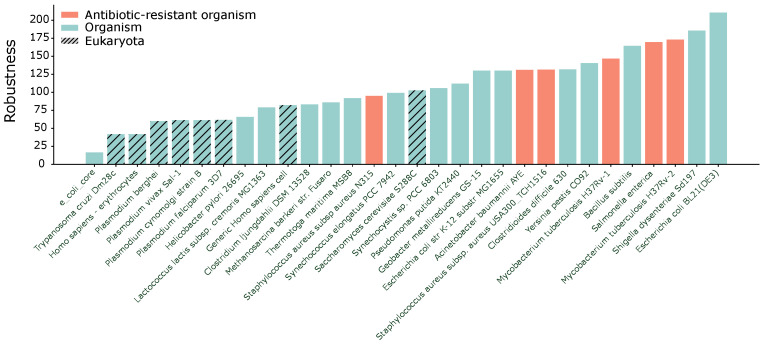
The robustness measured as the natural connectivity ${\bar{\lambda }}^{V}$ of 30 different BiGG models. The organisms resistant to antibiotics are shown in different colors. The models are ordered with increasing robustness.

**Figure 5 entropy-25-01537-f005:**
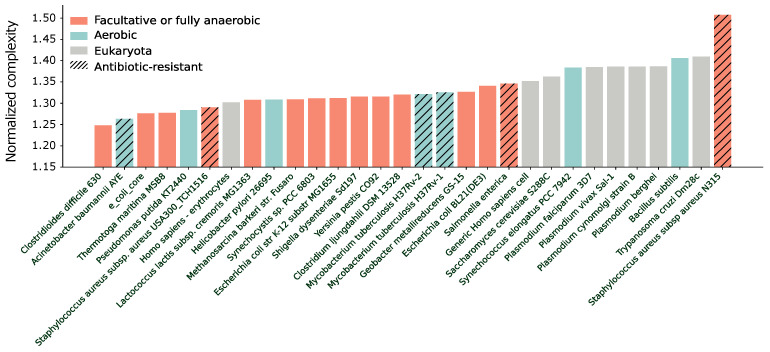
The complexity measured as the average search information $\sigma^{V}=\frac{S^{V}}{\log_{2}N}$ of 30 different BiGG models. The models are ordered with increasing complexity, and the y axis is zoomed in for visualization purposes.

## Data Availability

All data are publicly available on the BiGG models [49] web page in different formats. In this analysis, the *.json* format is used.

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
