# Peer review of "Robustness and Complexity of Directed and Weighted Metabolic Hypergraphs"

_entropy, 2023, doi:10.3390/e25111537_

Round 1
Reviewer 1 Report
Comments and Suggestions for Authors
In the presented manuscript the authors introduce a novel framework for metabolic network representation based on hyper-graphs. It introduces metrics to assess robustness and complexity, explores network directionality's effects, and compares various metabolic models, shedding light on antibiotic resistance and organism classification. This work presents an interesting methodology with high potential impact in the field. However, in my opinioin, in its current form, some changes are required for its publication. Following some observations and recommendations that I consider that should be taken into account:
Major considerations
- The introduction solely mentions stochastic modeling, overlooking equally, if not more, widely used methods such as kinetic metabolic models. This approach necessitates kinetic parameters, a key limitation absent in the presented approach. Another commonly employed method for analyzing the same large-scale metabolic network models explored with directed and weighted metabolic hypergraphs is constraint-based metabolic modeling. Its primary advantage over kinetic models lies in not mandating the definition of kinetic mechanisms or the use of kinetic parameters. In this context, this method shares the same benefit as the proposed approach concerning the absence of a requirement for kinetic parameters. Taking this into account, I would recommend the authors to discuss why the proposed method serves as a valid alternative or complement to the other methods mentioned.
- Section 2.4 raises the question of why some reactions in the original model have both lower and upper boundaries set to zero. Typically, a reaction being blocked by default might signify an error in the model. From a biological perspective, this has significant implications. An inactive reaction implies that the corresponding enzyme is never present in the organism, and the gene encoding this enzyme remains inactive. It's crucial to differentiate between reactions that are inherently blocked and those determined to be blocked through analyses such as FVA. If the blocked reactions are identified via FVA or similar methods, it should be explicitly stated in the manuscript. If not, the analysis should be performed with relaxed reaction boundaries, as this situation could potentially introduce errors into the metabolic network analysis.
- In the same section, it is mentioned that hyperedges with empty tails or heads correspond to sink reactions. However, it's important to note that reactions with empty heads or tails don't necessarily indicate sink or exchange reactions unless dead-end metabolites have been removed beforehand. This aspect requires addressing in the discussion to ensure clarity and accuracy in the model analysis.
Minor considerations
- In section 2.4 it is mentioned that hypergraphs are taken from BiGG Database, however it would be good to specify that they were generated from the S matrix of the models stored in the BiGG database
- At the end of section 3.1, it is noted that sink reactions are excluded from the analysis. Have the authors considered whether this limitation could be addressed by introducing a boundary node specifically for such analyses? This addition might offer the potential to incorporate factors like media composition, metabolite consumption/secretion, or intracellular pool dynamics into the analysis. While this could extend beyond the scope of the current work, it might be an intriguing topic to explore in the discussion.
Comments on the Quality of English Language- While the text is comprehensible, it would benefit from a review of its style and language.
Reviewer 2 Report
Comments and Suggestions for Authors
This manuscript provides fresh insights and methodologies in the realm of metabolic hypergraphs. The objective of the study is articulately stated, and the paper is well-structured and reader-friendly. Specific commendable aspects of the study are highlighted below:
Advantages
- Incorporation of Stoichiometric Coefficients: The novel approach of integrating stoichiometric coefficients effectively captures biological and chemical constraints within the hypergraph framework.
- Selection of Datasets: The curated choice of 30 metabolic networks from the BiGG Database offers a rich diversity, encompassing a range of organisms.
- Directionality of Reactions: The adoption of "lower bound" and "upper bound" parameters astutely addresses the directionality and potential reversibility of reactions.
Drawbacks:
To elevate the manuscript's quality, addressing the following concerns is essential. The authors are encouraged to revise in light of these suggestions:
- Lack of Citations for Prior Research: The manuscript appears to under-reference existing studies on the representation of reactions in hypergraphs. Incorporating relevant citations is strongly advised. For example, the authors can consider the following papers:
- Hypergraphs and cellular networks (doi:10.1371/journal.pcbi.10003859).
- Estimation of the number of extreme pathways for metabolic networks (doi:10.1186/1471-2105-8-363)
- Robustness of metabolic networks: A review of existing definitions (10.1016/j.biosystems.2011.06.002)
- Influence of Network Size: There are concerns that the introduced the network measurements related to robustness and complexity (e.g., communicability and search information) could be disproportionately affected by simple network parameters (numbers of metabolites, reactions, and hyperedges). This necessitates a more comprehensive analysis or clarification. For example, the authors may need to show no correlation between the network measurements and simple network parameters.
- Definition of Sinks and Sources: The criteria and rationale for determining sinks and sources remain somewhat ambiguous. A thorough clarification on this is suggested. Did the authors simply use the definitions of sources and sinks in BiGG models?
- Utilization of the BiGG Database: From the 138 networks available in the BiGG Database, the study examines only 30. Expanding the sample by analyzing more networks would arguably add robustness to the findings.
Moderate editing of English language required.
Round 2
Reviewer 1 Report
Comments and Suggestions for Authors
The authors have either addressed or justified all my comments, so in my opinion the manuscript can be published in its current form
Reviewer 2 Report
Comments and Suggestions for Authors
The authors have carefully revised the manuscript in accordance with the reviewer's comments and suggestions. As a result, the manuscript now meets the criteria for publication.